# OpenReview forum: "3D Interaction Geometric Pre-training for Molecular Relational Learning"
_ICLR.cc/2025/Conference — ICLR 2025 Conference Withdrawn Submission_

### Official Review · Reviewer_Navb · 2024-10-31

**Soundness:** 2
**Presentation:** 2
**Contribution:** 1
**Rating:** 3
**Confidence:** 5

**Summary:**

This paper designs a pre-training framework for Molecular Relational Learning. Including two self-supervised tasks including interaction geometry contrastive learning and intermolecular force prediction.

**Strengths:**

1. The paper is well-written and easy to understand.
2. The downstream tasks are diverse, and the performance improvements are substantial.

**Weaknesses:**

1. The molecular interaction environments are complex; simply moving two different molecular positions together with some random noise to simulate such a complex environment is far from realistic.
2. The intermolecular force prediction tasks based on this virtual environment are somewhat artificial. The pseudo force labels derived from the relative positions of two atoms are nonsensical.
3. What is the rationale behind the task of interaction geometry contrastive learning, and what type of interaction information does this task aim to learn?
4. How are the molecular pairs selected from the dataset, what kind of relation between these two molecules? The paper should provide more details regarding the validity of the simulation.

**Questions:**

See weakness

---

### Official Review · Reviewer_nno9 · 2024-10-31

**Soundness:** 3
**Presentation:** 3
**Contribution:** 3
**Rating:** 5
**Confidence:** 4

**Summary:**

This paper proposes a method for constructing a 3D virtual interaction environment, integrating it with a 2D representation through a contrastive learning paradigm to enable the 2D model to learn 3D geometric information of molecular interactions. Additionally, inter-molecular forces are used as a prediction target, supporting a deeper understanding of various molecular interaction processes. These two strategies significantly enhance the performance of the 2D model.

**Strengths:**

1. The proposed virtual interaction environment is a novel approach.
2. Experimental results demonstrate that 3DMRL effectively improves performance across various 2D MRL models.
3. The writing is clear and well-structured.

**Weaknesses:**

The authors claim that the proposed virtual interaction geometry appropriately reflects the real-world physics of molecular interactions. However, I find that the current implementation conflicts with established chemical knowledge in several ways:

1. Number of Interacting Molecules: The number of small molecules in the interaction environment should be variable, depending on the specific molecular properties of $g_{3D}^1$ and $g_{3D}^2$. However, the authors use a fixed $n = 5$, which assumes that all large molecules $g_{3D}^1$ interact with exactly five $g_{3D}^2$ molecules. This approach lacks scientific basis.

2. Heuristic Construction of Virtual Interaction Geometry: The method for constructing the virtual interaction geometry is entirely heuristic. It involves randomly selecting $n$ atoms from the non-aromatic ring of $g_{3D}^1$ and positioning $n$ smaller molecules $g_{3D}^2$ near these atoms by translating and rotating the original 3D coordinates $R^2$ of the smaller molecule. However, interactions between two molecules are complex, and assuming that interactions only occur in non-aromatic regions is inaccurate. While aromatic rings are generally stable, they can still participate in interactions, and randomly selected atoms from the non-aromatic ring may not necessarily engage in molecular interactions. Furthermore, the description in “Step 2: Positioning the Smaller Molecules” seems to lack chemical justification.

3. Pseudo Force Definition: Since actual molecular forces $f_k^i$ are not available, the authors propose using the direction between the $k$-th atom of the $i$-th small molecule and the $i$-th atom of the larger molecule to which the small molecule is attached as a pseudo force, defined as $f_k^i = R_i^1 / ||R_k^{2,i} - R_i^1||$. However, molecular interactions can involve multiple forces, such as hydrogen bonds, van der Waals forces, and ion-dipole interactions, each varying significantly in magnitude. Consequently, it is questionable whether $f_k^i = R_i^1 / ||R_k^{2,i} - R_i^1||$ can accurately represent real interaction forces.

The core concern is that if the proposed virtual interaction geometry and pseudo force do not align with chemical principles, it raises doubts about the effectiveness of the proposed method.

**Questions:**

1. Virtual Interaction Geometry as Data Augmentation: As noted in the "weakness" section, I believe the virtual interaction geometry largely lacks chemical validity. I am inclined to view the virtual interaction geometry as a form of data augmentation, where $g_{3D}^1$ and $g_{3D}^2$ are combined to create diverse configurations. Given that there are $n$ instances of $g_{3D}^2$, and their positions vary in each epoch, this can be seen as augmenting the data for $g_{3D}^1$ and $g_{3D}^2$. Following this view, an alternative approach could involve combining $g_{3D}^1$ and $g_{3D}^2$ while randomly perturbing atom coordinates, adding or removing atoms, or even creating combinations entirely at random instead of using the author’s proposed Step 2. If these variations are also effective, then it may be reasonable to interpret the virtual interaction geometry as data augmentation (rather than as a chemically meaningful virtual environment as the authors claim). Conversely, if these augmentations differ in key ways from the proposed method, it would be helpful to clarify the specific factors that make the proposed method effective.

2. Choice of $\mathbf{z}{2d}$ over $\mathbf{z}{3d}$ in Experiments: In the experimental section, the authors appear to have used only $\mathbf{z}{2d}$ for the experiments. Why was $\mathbf{z}{3d}$ not selected? Since $\mathbf{z}{2d}$ learns 3D information through contrastive learning with $\mathbf{z}{3d}$, $\mathbf{z}{3d}$ should, in theory, also acquire topological information from $\mathbf{z}{2d}$. It would be interesting to see whether $\mathbf{z}{3d}$ obtained via contrastive learning improves performance compared to the direct application of $R{vr}$ to the 3D encoder to derive $\mathbf{z}_{3d}$.

3. Pre-training Baselines in Table 2: In Table 2, the authors compare the performance of the CIGIN model on molecular interaction tasks using different pre-training strategies, with 3D Infomax, GraphMVP, and MoleculeSDE chosen as baselines. These models integrate 2D graph and 3D information in their training. I believe it would be beneficial to include purely 3D-based methods, such as Uni-mol, as additional baselines.

---

### Official Review · Reviewer_uMgd · 2024-11-01

**Soundness:** 3
**Presentation:** 3
**Contribution:** 2
**Rating:** 3
**Confidence:** 4

**Summary:**

This paper introduces 3DMRL, a molecular pre-training framework that leverages 3D interaction geometries to enhance 2D models for molecular representation learning (MRL). To efficiently obtain large-scale 3D interaction data, this paper proposes a strategy that constructs virtual interaction geometries by positioning small molecules around a central larger molecule. Using this dataset, a 2D-3D contrastive learning task and an intermolecular force prediction task is combined for pre-training. The results demonstrate the superiority and generalizability of the proposed method.

**Strengths:**

1. This paper innovatively introduces an interaction-based perspective for MRL pre-training.
2. The virtual interaction dataset construction is efficient and presents a clear methodology, making it easily reproducible.

**Weaknesses:**

1. Compared to prior methods like GraphMVP and 3D Infomax, which also leverage contrastive tasks between 2D and 3D graphs, this paper’s 'only cont.' variant outperforms the contrastive baselines, as shown in Tables 2, 3, and Figure 3(a). Could the authors elaborate on the primary distinctions between this variant and existing methods?
2. For the force prediction task, directly using the vector direction from each atom in the small molecule to the corresponding atom in the larger molecule lacks clear theoretical or motivational support. Moreover, as indicated by Figure 3, this task contributes only a marginal performance gain.
3. Certain notations are unclear. For instance, in Line 307, it is puzzling why the output of the 'Projection' function is in $\mathbb{R}^d$ instead of $\mathbb{R}^3$.

**Questions:**

See the weaknesses above.

---

### Official Review · Reviewer_KyD6 · 2024-11-02

**Soundness:** 2
**Presentation:** 3
**Contribution:** 2
**Rating:** 5
**Confidence:** 5

**Summary:**

The paper presents a novel 3D interaction geometry pre-trained method for molecule relational learning, where fine-grained molecule interaction is learned through force prediction loss. Experiments on six popular datasets demonstrate the effectiveness of the proposed method based on molecule relational learning.

**Strengths:**

1. The introduction of 3D information pre-training in molecule relational learning is innovative.
2. The model obtains excellent results on a variety of datasets showing its generality.

**Weaknesses:**

1. The major concern is that intermolecular interactions and chemical bonds are distinct, and Local frame is suitable only for single molecule modeling. On the other hand, the 3D GNN framework, which considers cutoff distances, may truncate long-range forces and does not differentiate between intermolecular forces and chemical bonds in modeling molecular pairs.
2. In contrastive pre-training, Line 247-248, aligning multiple 3D solvent representations to the same 2D representation introduces ambiguity, making it challenging for the model to learn a consistent mapping. This bias can result in feature alignment that deviates from true chemical and physical properties.
3. Acquiring 3D structures is costly, and those generated by common force fields algorithms may be unsuitable for solvated molecular.

**Questions:**

1. How is the initial position obtained?
2. Some datasets do not contain energy labels, such as the Chromophore dataset and DDI dataset, how to obtain the force and ground true force label?
3. What is the ratio of large molecules to small molecules? Does the ratio of different molecules have an effect on the results?

---

### Note · Authors · 2024-11-25

I have read and agree with the venue's withdrawal policy on behalf of myself and my co-authors.